# GSK3β regulates AKT-induced central nervous system axon regeneration via an eIF2Bε-dependent, mTORC1-independent pathway

Xinzheng Guo[1], William D Snider[2], Bo Chen[1,3]*

[1]Department of Ophthalmology and Visual Science, Yale University School of Medicine, New Haven, United States; [2]UNC Neuroscience Center, University of North Carolina at Chapel Hill, Chapel Hill, United States; [3]Department of Neurobiology, Yale University School of Medicine, New Haven, United States

**Abstract** Axons fail to regenerate after central nervous system (CNS) injury. Modulation of the PTEN/mTORC1 pathway in retinal ganglion cells (RGCs) promotes axon regeneration after optic nerve injury. Here, we report that AKT activation, downstream of *Pten* deletion, promotes axon regeneration and RGC survival. We further demonstrate that GSK3β plays an indispensable role in mediating AKT-induced axon regeneration. Deletion or inactivation of GSK3β promotes axon regeneration independently of the mTORC1 pathway, whereas constitutive activation of GSK3β reduces AKT-induced axon regeneration. Importantly, we have identified eIF2Bε as a novel downstream effector of GSK3β in regulating axon regeneration. Inactivation of eIF2Bε reduces both GSK3β and AKT-mediated effects on axon regeneration. Constitutive activation of eIF2Bε is sufficient to promote axon regeneration. Our results reveal a key role of the AKT-GSK3β-eIF2Bε signaling module in regulating axon regeneration in the adult mammalian CNS.

*For correspondence: b.chen@yale.edu

Competing interests: The authors declare that no competing interests exist.

## Introduction

Axon injury in the adult mammalian central nervous system (CNS) causes irreversible damages and permanent loss of functions due to a diminished intrinsic regenerative capability of the mature CNS neurons as well as an inhibitory extrinsic environment (*Horner and Gage, 2000*; *Yiu and He, 2006*). Reactivation of the intrinsic regenerative capability promotes CNS axon regeneration after injury (*Fischer et al., 2004*; *Gaub et al., 2011*; *Liu et al., 2010*; *Moore et al., 2009*; *O'Donovan et al., 2014*; *Park et al., 2008*; *Smith et al., 2009*; *Watkins et al., 2013*). Significantly, deletion of PTEN (phosphatase and tensin homolog) in adult retinal ganglion cells (RGCs) promotes axon regeneration after optic nerve injury (*Park et al., 2008*). Loss of PTEN leads to the accumulation of $PIP_3$ (phosphatidylinositol-3,4,5-trisphosphate), resulting in the activation of the serine/threonine kinase AKT via PDK1-mediated phosphorylation (*Carnero et al., 2008*; *Luo et al., 2003*). AKT is a critical node in cell signaling downstream of growth factors, cytokines, and other cellular stimuli and regulates a wide spectrum of cellular functions, which include cell survival, growth, proliferation, metabolism, and migration (*Manning and Cantley, 2007*). The role of AKT in CNS axon regeneration remains to be revealed.

Activation of mTOR complex 1 (mTORC1) plays an important role in mediating *Pten* deletion-induced axon regeneration (*Park et al., 2008*). However, mTORC1-independent pathway(s) may exist to regulate axon regrowth in the *Pten*-deficient neurons because 1) pharmacological inhibition of mTORC1 by rapamycin treatment only partially neutralizes the effects of *Pten* deficiency on axon

**eLife digest** The central nervous system consists of the neurons that make up the brain, retina, and spinal cord. Neurons transmit electrical signals along a cable-like structure called an axon. However, an axon cannot regenerate itself, and so injuries that crush or sever the axons can lead to permanent damage. This happens for two reasons: neurons don't have the same regenerative ability as other cells, and the environment in the central nervous system restricts cell growth.

The optic nerve transmits visual information from the eye to the brain. Studies in mice with a damaged optic nerve show that it is possible to regenerate the axons of neurons that lack a protein known as PTEN. These studies revealed one molecular pathway by which eliminating PTEN helps to boost the regrowth of axons.

Now, Guo et al. identify another independent pathway by which eliminating PTEN helps promote axon regeneration in damaged mouse optic nerves. This pathway starts with a growth-promoting enzyme called AKT, which is turned on in neurons that lack PTEN. Indeed, injecting mice with an active form of this enzyme caused the optic nerve fiber to regrow in mice whose optic nerve had been crushed.

Further experiments revealed that AKT activates a pathway in which another enzyme called GSK3β acts on a protein called eIF2Bε. A future challenge is to simultaneously manipulate the different signaling pathways that have been linked to axon regrowth to investigate whether this combined approach could help repair damage to the central nervous system.

regeneration (*Park et al., 2008*); 2) genetic ablation of TSC1 (tuberous sclerosis complex 1), a negative regulator of mTORC1, or manipulation of downstream effectors of mTORC1 only partially but not completely recapitulated the axon regeneration effects mediated by *Pten* deletion (*Park et al., 2008*; *Yang et al., 2014*).

GSK3 (glycogen synthase kinase 3) is a signal transducer of AKT. In mammals, the GSK3 family consists of two members, GSK3α and GSK3β. AKT inactivates the kinase activity of GSK3 through phosphorylation of GSK3α at Ser21 or GSK3β at Ser9 (*Cross et al., 1995*). Studies in the peripheral nervous system (PNS) have generated different results regarding the role of GSK3 in regulating PNS axon regeneration. Using *Gsk3a*-S21A/*Gsk3b*-S9A double knock-in mice, in which GSK3α/GSK3β cannot be inactivated by AKT-mediated phosphorylation, one group did not observe obviously phenotype in sensory axon regeneration (*Zhang et al., 2014*), whereas the other group reported that sustained GSK3 activity markedly facilitated sciatic nerve regeneration (*Gobrecht et al., 2014*). In the CNS, pharmacological inactivation of GSK3 by the administration of lithium, a GSK3 inhibitor, stimulates axon formation and elongation in the presence or absence of inhibitory substrates after spinal cord injury (*Dill et al., 2008*). However, the mechanism of lithium action is not completely clear as many other potential targets including a number of vital enzymes are also inhibited by lithium in an uncompetitive manner (*Phiel and Klein, 2001*). Thus, genetic evidence is needed to examine the role of GSK3 in CNS axon regeneration.

In the present study using optic nerve crush (ONC) to model CNS axon injury, we delineate the role of GSK3β in regulating AKT-induced axon regeneration. We further identified eIF2Bε (eukaryotic translation initiation factor 2B epsilon subunit), a substrate of GSK3β, as a novel factor to regulate both GSK3β and AKT-mediated effects on axon regeneration.

## Results

### AKT activation, downstream of *Pten* deletion, promotes axon regeneration

AKT activation, indicated by phosphorylation at Ser473 (*Bozulic and Hemmings, 2009*), was readily detectable in RGCs labeled by anti-Tuj1 immunoreactivity in the developing retina and was significantly reduced in adult mice (*Figure 1—figure supplement 1A–B*). PTEN negatively regulates the PI3K/AKT signaling pathway. To examine whether the deletion of *Pten* leads to AKT activation, we injected AAV-Cre into the vitreous body of adult *Pten*^f/f^ mice, resulting in Cre-mediated *Pten*

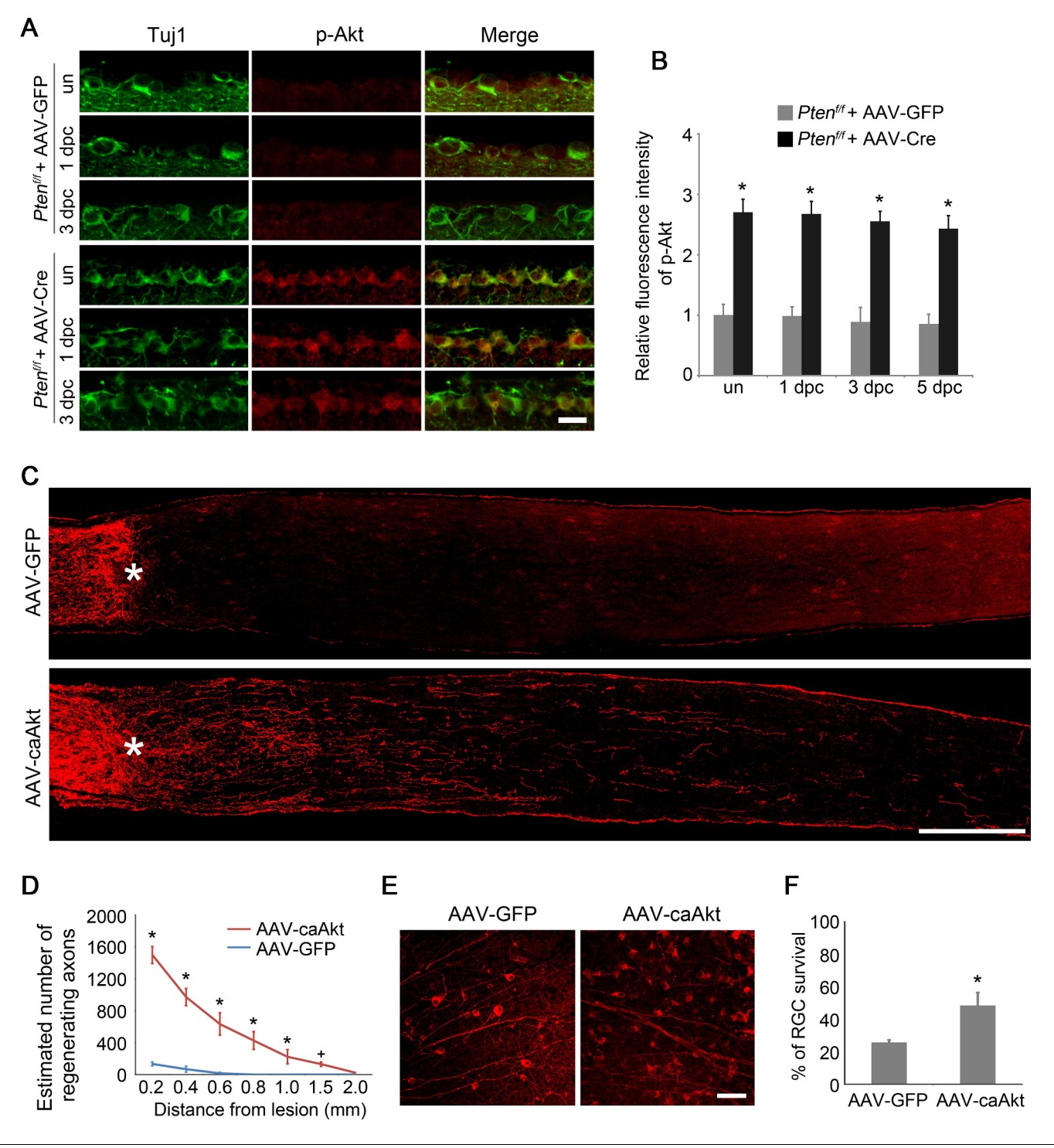

**Figure 1.** AKT activation promotes axon regeneration and RGC survival. (**A**) Detection of AKT Phosphorylation at Ser473 in *Pten*-deleted RGCs labeled by anti-Tuj1 immunohistochemistry, with or without (un) optic nerve injury. Scale bar, 20 μm. (**B**) Quantification of phospho-AKT immunofluorescence intensity following *Pten* deletion. Data are presented as mean ± s.d., n=4 per group. *p<0.001, Two-way ANOVA with Bonferroni *post hoc* test. (**C**) Confocal images of optic nerve sections showing regenerating axons labeled by anti-GAP43 immunohistochemistry at 2 weeks after optic nerve injury from AAV-GFP or AAV-caAkt injected eyes. *, crush site. Scale bar, 200 μm. (**D**) Quantification of regenerating axons at different distances distal to the lesion site. Data are presented as mean ± s.d., n=5 per group. *p<0.01, +p<0.05, Two-way ANOVA with Bonferroni *post hoc* test. (**E**) Confocal images of retinal whole-mounts showing surviving Tuj1+ RGCs at 2 weeks after optic nerve injury. Scale bar, 50 μm. (**F**) Quantification of RGC survival at 2 weeks

*Figure 1 continued on next page*

*Figure 1 continued*

after injury, expressed as a percentage of total Tuj1$^+$ RGCs in the uninjured retina. Data are presented as mean ± s.d., n=5 per group. *p<0.01, Student's *t* test.

The following figure supplements are available for figure 1:

**Figure supplement 1.** Developmental analysis of AKT phosphorylation in RGCs.

**Figure supplement 2.** Inhibition of mTORC1 partially reduces AKT-induced axon regeneration.

deletion in RGCs (*Figure 1—figure supplement 1C–D*), (*Park et al., 2008*). Two weeks after viral injection, we detected increased phosphorylation of AKT at Ser473 in RGCs either with or without ONC, in comparison with *Pten*$^{f/f}$mice injected with AAV-GFP as a control (*Figure 1A–B*). To investigate whether AKT activation is sufficient to promote axon regeneration, we injected AAV-caAKT (a constitutively active form of AKT [*Kohn et al., 1996*]), or AAV-GFP as a control, into the vitreous body of adult wild-type mice (*Figure 1—figure supplement 1E–F*) and performed ONC two weeks after viral injection. Axon regeneration was examined two weeks after ONC using immunohistochemistry for GAP-43 (*Koprivica et al., 2005*; *Leon et al., 2000*). While very few axon fibers extended beyond the crush site in the AAV-GFP injected control, robust axon regeneration was stimulated in the AAV-caAKT injected retina (*Figure 1C*). Upon AKT activation, many regenerating axon fibers were observed at the proximal region of the crush site. The number of regenerating axons gradually declined over a longer distance from the lesion site (*Figure 1D*). To assess the effect of AKT activation on RGC survival, we quantified the number of Tuj1$^+$ cells using immunohistochemistry on retinal flatmount preparations two weeks after ONC (*Figure 1E*). In comparison to 25.7% of RGCs remaining in the AAV-GFP injected control, significantly more RGCs (48.4%) were scored in the AAV-caAKT injected retina (*Figure 1F*). Our results indicate that *Pten* deletion activates AKT, and AKT activation promotes both axon regeneration and RGC survival after optic nerve injury.

## GSK3β inactivation mediates AKT-induced axon regeneration

The mTORC1 pathway plays an important role in mediating *Pten* deletion-induced axon regeneration. To examine whether blockade of mTORC1 signaling also affects AKT-induced axon regeneration, we administered rapamycin to inhibit mTORC1 in the AAV-caAKT-injected retinas. In comparison with the vehicle treatment as a control, rapamycin treatment partially reduced the number of regenerating axon fibers (*Figure 1—figure supplement 2A–B*). The residual regenerative effect could be due to incomplete blockage of mTORC1 signaling by the drug treatment, or additional mTORC1-independent pathway(s) may exist to regulate axon regeneration downstream of AKT. We also assessed RGC survival after rapamycin treatment. AKT-induced RGC survival was largely neutralized by rapamycin treatment (*Figure 1—figure supplement 2C–D*).

Given the prominent role of GSK3 in developmental axon extension (*Zhou et al., 2004*) and implication of GSK3 in the regenerative axon growth in the PNS (*Gobrecht et al., 2014*; *Saijilafu et al., 2013*), we next examined the role of GSK3 in CNS axon regeneration. Phosphorylation of GSK3β at Ser9 results in inactivation of its kinase activity (*Cross et al., 1995*). Phospho-GSK3β level was higher in the developing retina relative to the adult age (*Figure 2—figure supplement 1*). Two weeks after AAV-caAKT injection in adult wild-type mice, we detected increased phosphorylation of GSK3β at Ser9 in RGCs either with or without ONC, in comparison with the AAV-GFP injected retina as a control (*Figure 2A–B*). We also examined whether *Pten* deletion also led to GSK3β phosphorylation at Ser9 in RGCs. As expected, increased phosphorylation of GSK3β was observed in *Pten*-deficient RGCs (*Figure 2—figure supplement 2*). Our results indicate that GSK3β is a downstream target of PTEN/AKT signaling in RGCs. To investigate whether GSK3β inactivation mediates AKT-induced axon regeneration and RGC survival, we co-injected AAV-caAKT with AAV-GSK3β S9A, a Ser-to-Ala mutant of GSK3β that cannot be phosphorylated by AKT and therefore is constitutively active (*Eldar-Finkelman et al., 1996*). In comparison with the AAV-caAKT and AAV-GFP co-injection, regenerative axon growth was significantly reduced by AAV-GSK3β S9A co-injection (*Figure 2C–D*). However, AKT-induced RGC survival was not affected by the expression of GSK3β S9A (*Figure 2E–*

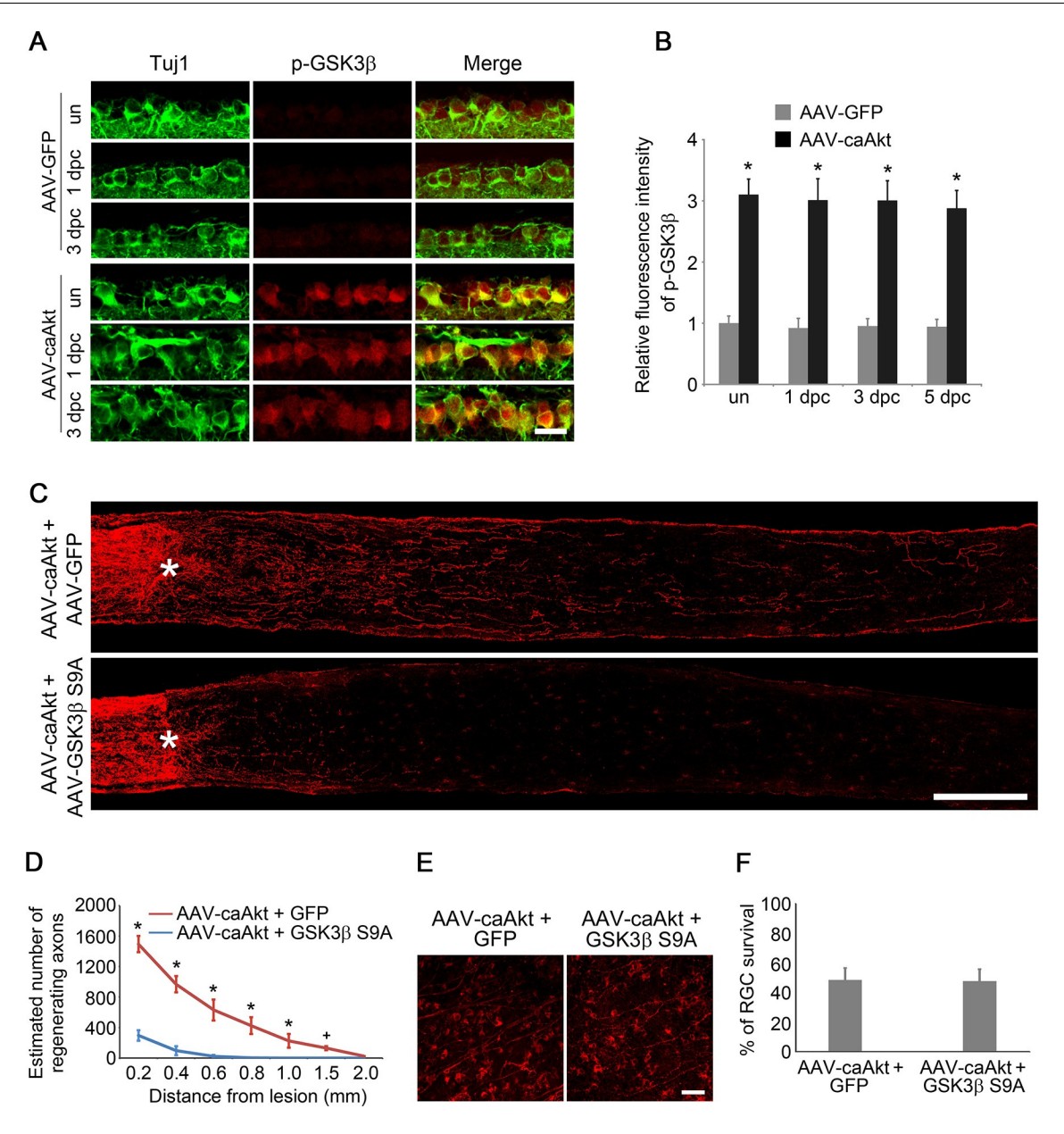

**Figure 2.** GSK3$\beta$ is an essential downstream effector to mediate AKT-induced axon regeneration. (**A**) Immunohistochemical detection of GSK3$\beta$ phosphorylation at Ser9 in retinal sections from AAV-GFP or AAV-caAkt-injected eyes, either with or without optic nerve injury. Scale bar, 20 μm. (**B**) Quantification of phospho-GSK3$\beta$ immunofluorescence intensity. Data are presented as mean ± s.d., n=4 per group. *$p<0.001$, Two-way ANOVA with Bonferroni *post hoc* test. (**C**) Confocal images of optic nerve sections showing regenerating axons labeled by anti-GAP43 immunohistochemistry at 2 weeks after optic nerve injury from AAV-caAKT + AAV-GFP or AAV-caAkt + AAV-GSK3$\beta$ S9A injected eyes. *, crush site. Scale bar, 200 μm. (**D**) Quantification of regenerating axons at different distances distal to the lesion site. Data are presented as mean ± s.d., n=5 per group. *$p<0.01$, +$p<0.05$, Two-way ANOVA with Bonferroni *post hoc* test. (**E**) Confocal images of retinal whole-mounts showing surviving Tuj1[+] RGCs at 2 weeks after optic nerve injury. Scale bar, 50 μm. (**F**) Quantification of RGC survival at 2 weeks after injury, expressed as a percentage of total Tuj1[+] RGCs in the uninjured retina. Data are presented as mean ± s.d., n=5 per group.

The following figure supplements are available for figure 2:

**Figure supplement 1.** Developmental analysis of GSK3$\beta$ phosphorylation in RGCs.

**Figure supplement 2.** *Pten* deletion results in GSK3$\beta$ phosphorylation in RGCs.

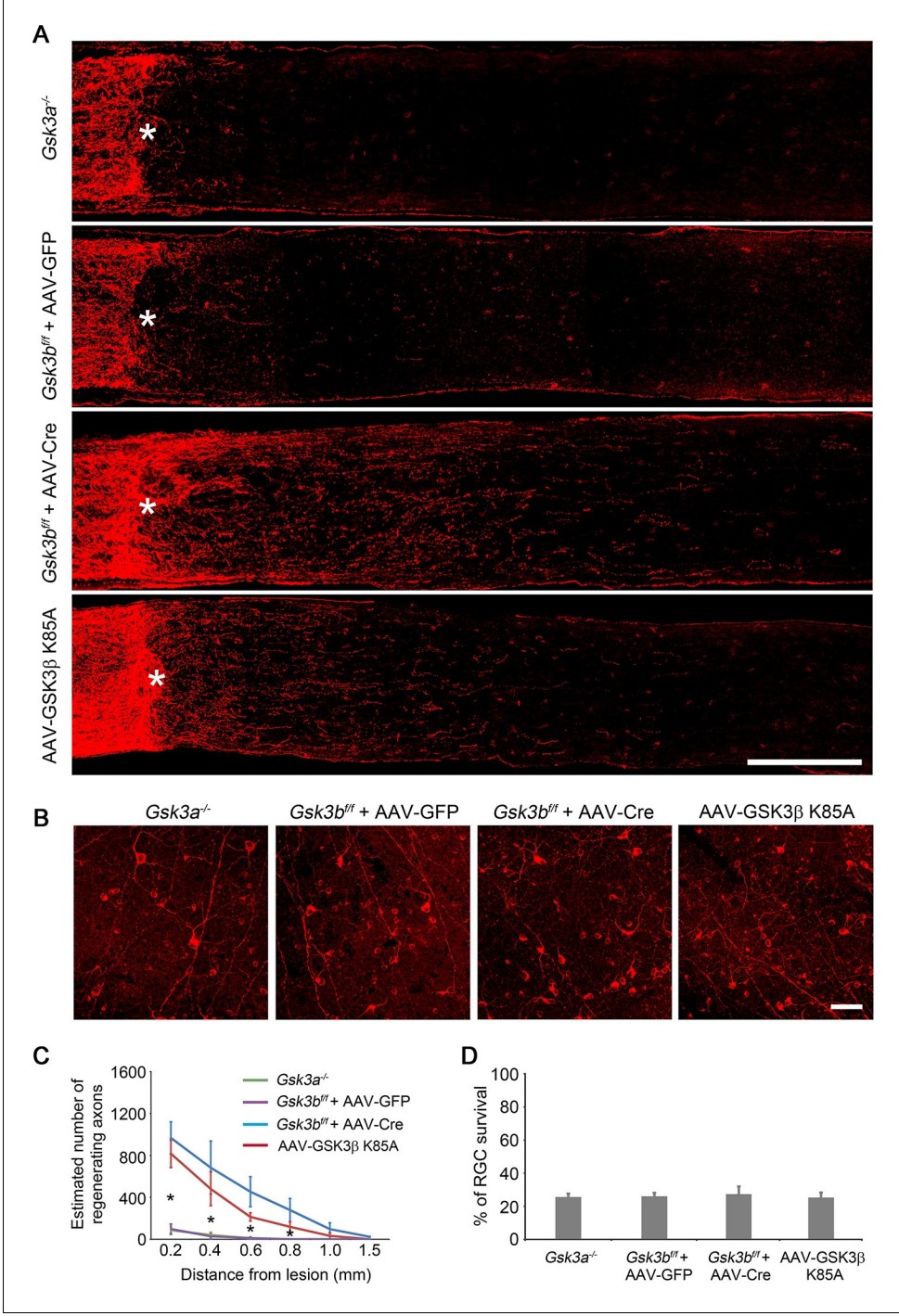

**Figure 3.** Deletion or inactivation of GSK3$\beta$ promotes axon regeneration. (**A**) Confocal images of optic nerve sections showing regenerating axons labeled by anti-GAP43 immunohistochemistry at 2 weeks after optic nerve crush from *Gsk3a*[−/−] mice, *Gsk3b*[f/f] mice injected with AAV-GFP or AAV-Cre, or AAV-GSK3$\beta$ K85A injected wild-type mice. *, crush site. Scale bar, 200 μm. (**B**) Confocal images of retinal whole-mounts showing surviving Tuj1[+] RGCs at 2 weeks after optic nerve injury. Scale bar, 50 μm. (**C**) Quantification of regenerating axons at different distances distal to the lesion site. Data are presented as mean ± s.d., n=5 per group. *p<0.01, Two-way ANOVA with Bonferroni *post hoc* test. (**D**) Quantification of RGC survival at 2 weeks after injury, expressed as a percentage of total Tuj1[+] RGCs in the uninjured retina. Data are presented as mean ± s.d., n=5 per group.

The following figure supplements are available for figure 3:

*Figure 3 continued on next page*

*Figure 3 continued*

**Figure supplement 1.** Cre-mediated *Gsk3b* deletion in RGCs.
**Figure supplement 2.** A time course study of axon regeneration in *Gsk3b*-deleted RGCs.
**Figure supplement 3.** AKT activation increases GSK3α phosphorylation.

*F*). Our results indicate that AKT phosphorylates and thus inactivates GSK3β, leading to improved axon regeneration, but not RGC survival.

## Deletion or inactivation of GSK3β is sufficient to promote axon regeneration

To investigate whether deletion of *Gsk3b* promotes RGC axon regeneration, we injected AAV-Cre, or AAV-GFP as a control, in the adult *Gsk3b^{f/f}* mice (*Patel et al., 2008*) (*Figure 3—figure supplement 1*), and examined axon regeneration two weeks after ONC. While no obvious regenerating axon fibers were observed in the AAV-GFP injected control, *Gsk3b* deletion resulted in improved axon regeneration (*Figure 3A*). We next examined the time course of axon regeneration in *Gsk3b*-deficient RGCs after injury (*Figure 3—figure supplement 2*). At 1 day after ONC, optic nerve fibers terminated at the crush site in *Gsk3b^{f/f}* mice injected with either AAV-GFP or AAV-Cre. At 3 days after ONC, we observed axon sprouting in the *Gsk3b*-deleted RGCs. At 7 days after ONC, regenerating axon fibers extended beyond the lesion site in *Gsk3b*-deficient RGCs, overcoming the inhibitory environment at the lesion site labeled by immunohistochemistry for chondroitin sulfate proteoglycan (CSPG) and glial fibrillary acidic protein (GFAP). However, deletion of *Gsk3a* did not stimulate the regenerative response of injured axons (*Figure 3A*), although AKT activation also increased its phosphorylation at Ser21 (*Figure 3—figure supplement 3*). To further examine whether elimination of the kinase activity of GSK3β is responsible for *Gsk3b* deletion-induced axon regeneration, we injected AAV-GSK3β K85A, a kinase dead mutant of GSK3β that inhibits endogenous GSK3β in a dominant negative manner (*Dominguez et al., 1995*), into the vitreous body of adult wild-type mice. Two weeks after ONC, we observed many regenerating axon fibers extending beyond the lesion site in the AAV-GSK3β K85A injected mice, a regenerative response slightly weaker in comparison with *Gsk3b* deletion (*Figure 3A,C*). We also analyzed neuronal survival two weeks after ONC in the retinas of *Gsk3a* deletion, *Gsk3b* deletion, and AAV-GSK3β K85A injection. None of these manipulations changed the rate of RGC survival (*Figure 3B,D*). Our results indicate that the deletion of *Gsk3b*, but not *Gsk3a*, promotes RGC axon regeneration likely through inactivation of the kinase activity of GSK3β.

## GSK3β acts independently of mTORC1 to regulate axon regeneration

Activation of the mTORC1 pathway is a well-established downstream signaling mechanism in *Pten* deletion-induced axon regeneration (*Park et al., 2008*; *Yang et al., 2014*). And our results demonstrate that GSK3β may represent another important signaling pathway downstream of PTEN/AKT to regulate CNS axon regrowth. To distinguish whether *Gsk3b* deletion-induced axon regeneration is through activation of the mTORC1 pathway or through an mTORC1-independent mechanism, we first examined whether mTORC1 signaling is altered by *Gsk3b* deletion using immunohistochemistry for phospho-S6 ribosomal protein (p-S6) to monitor the activity of mTORC1 in RGCs (*Hay and Sonenberg, 2004*; *Park et al., 2008*). While AKT activation significantly increased the mTORC1 activity, evidenced by a higher fluorescence intensity of p-S6 immunoreactivity in RGCs in adult wild-type mice injected with AAV-caAKT compared to the control injection, *Gsk3b* deletion did not perturb the activity of mTORC1 as the immunofluorescence intensity of p-S6 remained at the same low level as the control (*Figure 4A–B*). To further assess whether the mTORC1 pathway contributes to the regenerative effects induced by *Gsk3b* deletion, we administered rapamycin to inhibit mTORC1 activity in *Gsk3b*-deleted RGCs. While rapamycin markedly reduced AKT-induced axon regeneration and RGC survival (*Figure 1—figure supplement 2*), *Gsk3b* deficiency-induced axon regeneration was not affected by rapamycin treatment, and neither was RGC survival in *Gsk3b*-deleted RGCs (*Figure 4—figure supplement 1*). However, treatment with the protein synthesis inhibitor anisomycin

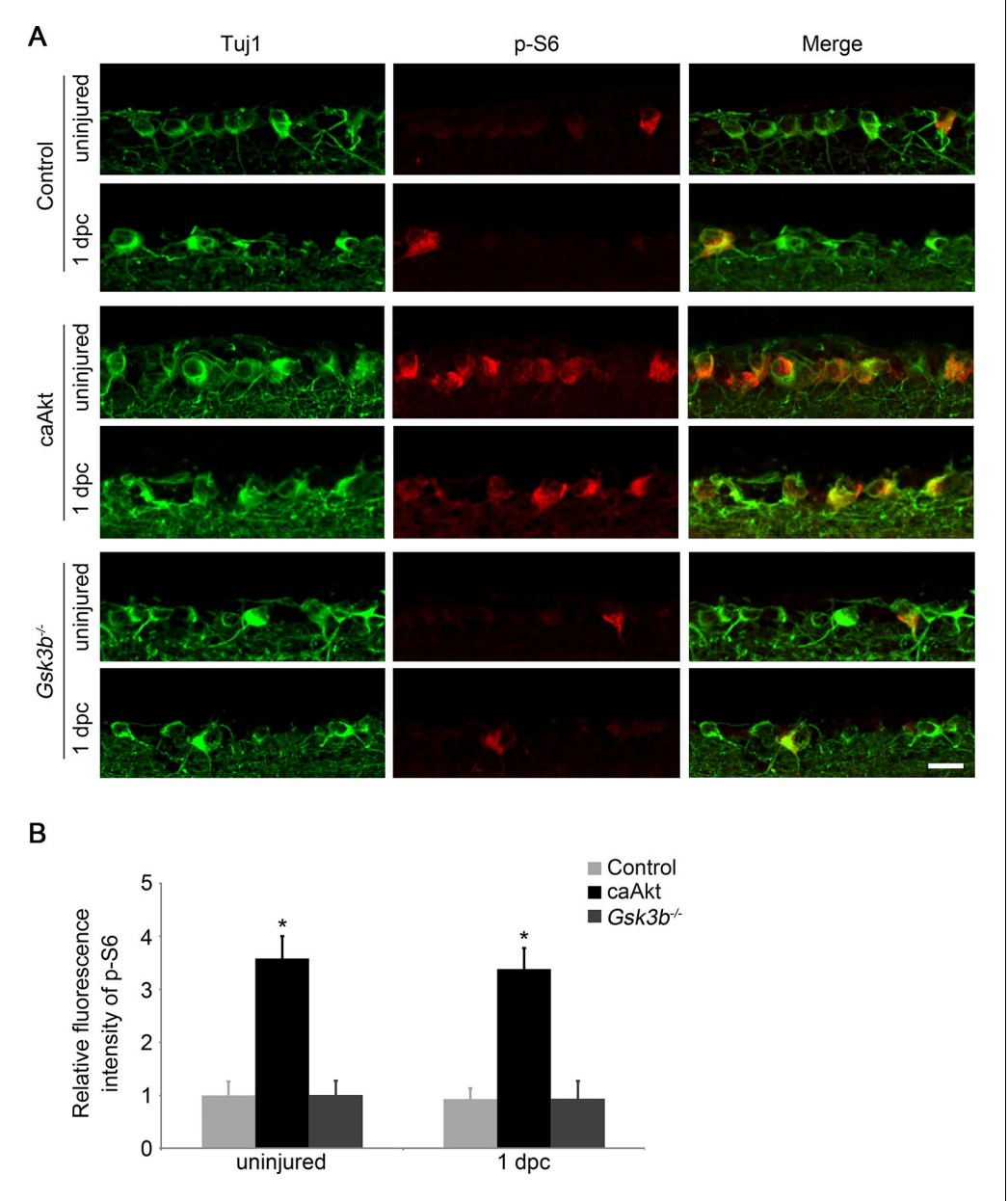

**Figure 4.** *Gsk3b* deletion does not alter the mTORC1 pathway. (**A**) Confocal images of retinal sections showing double immunolabeling for Tuj1[+] RGCs and phospho-S6 from AAV-GFP, AAV-caAkt, or AAV-Cre (*Gsk3b^{f/f}*)-injected eyes, with or without optic nerve injury. Scale bar, 20 μm. (**B**) Quantification of phospho-S6 immunofluorescence intensity. Data are presented as mean ± s.d., n=4 per group. *p<0.001, Two-way ANOVA with Bonferroni *post hoc* test.

The following figure supplement is available for figure 4:

**Figure supplement 1.** *Gsk3b* deletion-induced axon regeneration is sensitive to protein synthesis inhibition but not to mTORC1 inhibition.

significantly reduced *Gsk3b* deletion induced axon regeneration (*Figure 4—figure supplement 1*). Our results indicate that GSK3β-mediated effects on axon regeneration require protein synthesis but are achieved through an mTORC1-independent mechanism.

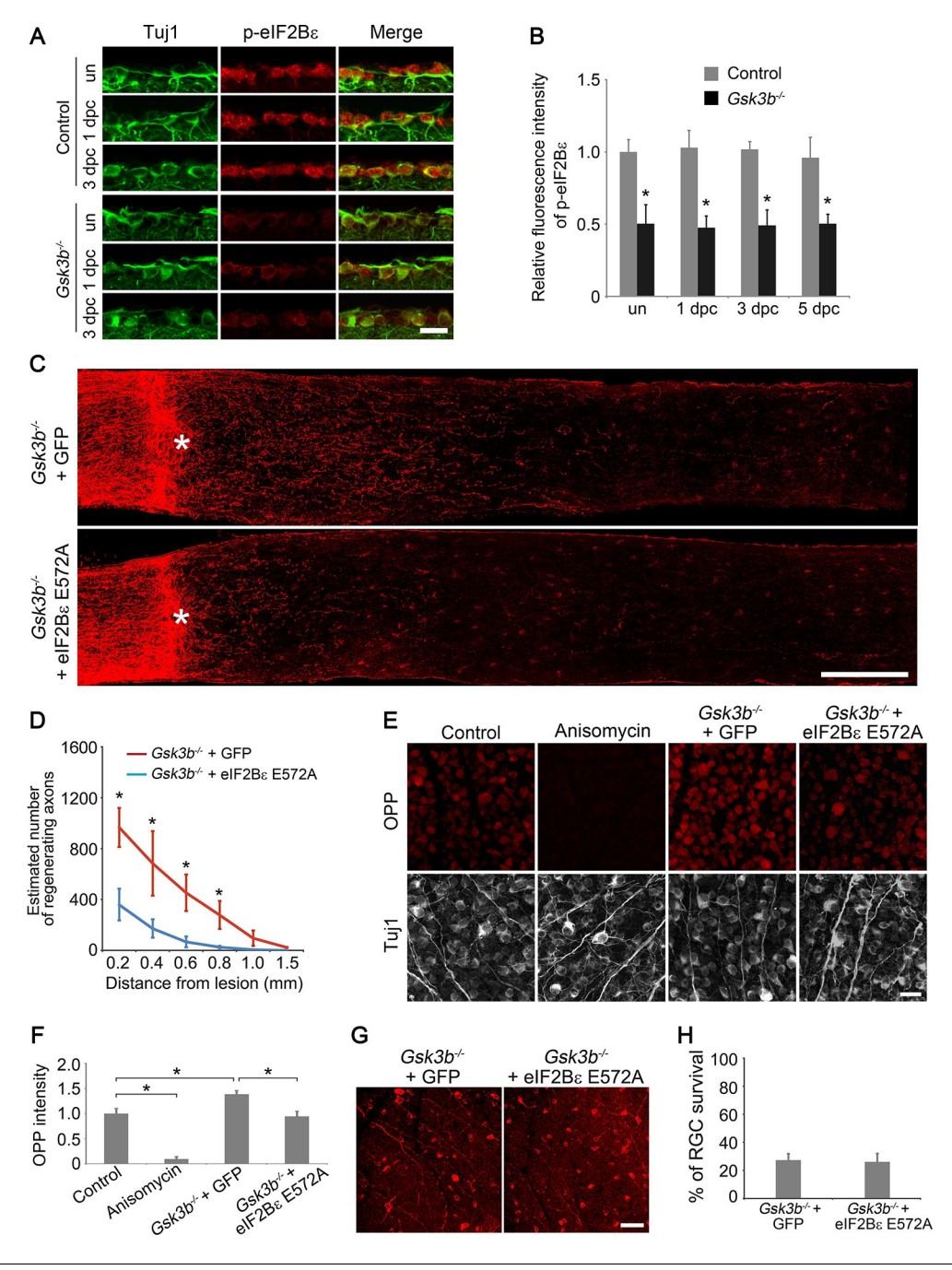

**Figure 5.** eIF2B$\varepsilon$ is required for *Gsk3b* deletion-induced axon regeneration. (**A**) Confocal images of retinal sections showing double immunolabeling for Tuj1$^+$ RGCs and phospho-eIF2B$\varepsilon$ from *Gsk3b$^{f/f}$* mice injected with AAV-GFP (Control) or AAV-Cre (*Gsk3b$^{-/-}$*), with or without optic nerve injury. Scale bar, 20 μm. (**B**) Quantification of phospho-eIF2B$\varepsilon$ immunofluorescence intensity. Data are presented as mean ± s.d., n=5 per group. *p<0.001, Two-way ANOVA with Bonferroni *post hoc* test. (**C**) Confocal images of optic nerve sections showing regenerating axons labeled by anti-GAP43 immunohistochemistry at 2 weeks after optic nerve crush from *Gsk3b$^{f/f}$* mice injected with AAV-Cre + AAV-GFP or AAV-Cre + AAV-eIF2B$\varepsilon$ E572A. *, crush site. Scale bar, 200 μm. (**D**) Quantification of regenerating axons from *Gsk3b$^{f/f}$* mice injected with AAV-Cre + AAV-GFP or AAV-Cre + AAV-eIF2B$\varepsilon$ E572A at different distances distal to the lesion site. Data are presented as mean ± s.d., n=9 per group. *p<0.01, Two-way ANOVA with Bonferroni *post hoc* test. (**E**) OPP Alexa Fluor 594 protein synthesis assay in retinal whole-mounts from *Gsk3b$^{f/f}$* mice injected with AAV-GFP (Control), AAV-Cre (*Gsk3b$^{-/-}$*) + AAV-GFP, or AAV-Cre + AAV-eIF2B$\varepsilon$ E572A or treated with anisomycin. Scale bar, 25 μm. (**F**) Quantification of OPP fluorescence intensity. Data are *Figure 5 continued on next page*

*Figure 5 continued*
presented as mean ± s.d., n=5 per group. *p<0.01, One-way ANOVA with Tukey's test. (**G**) Confocal images of retinal whole-mounts showing surviving Tuj1[+] RGCs at 2 weeks after optic nerve injury. Scale bar, 50 µm. (**H**) Quantification of RGC survival at 2 weeks after injury, expressed as a percentage of total Tuj1[+] RGCs in the uninjured retina. Data are presented as mean ± s.d., n=5 per group.
The following figure supplements are available for figure 5:
**Figure supplement 1.** Developmental analysis of eIF2Bε phosphorylation in RGCs.
**Figure supplement 2.** Analysis of eIF2Bε phosphorylation in *Gsk3a* knockout mice.

## eIF2Bε is required for GSK3β-mediated effects on axon regeneration

GSK3β regulates many cellular processes through phosphorylation of its diverse substrates that include transcription factors, cytoskeletal proteins, motor proteins, and proteins involved in the regulation of cell growth (*Liu et al., 2012*; *Saijilafu et al., 2013*; *Sutherland, 2011*). eIF2B (eukaryotic translation initiation factor 2B) is a heteropentameric guanine nucleotide exchange factor that converts eIF2 (eukaryotic translation initiation factor 2) from the inactive GDP–bound form to the active GTP-bound complex, representing a rate-limiting step in protein translation initiation (*Campbell et al., 2005*; *Pavitt, 2005*). eIF2Bε (eIF2B epsilon subunit), the largest catalytic subunit of eIF2B, is inhibited by GSK3β phosphorylation at the Ser535 site, linking regulation of global protein synthesis to GSK3β signaling (*Pap and Cooper, 2002*; *Wang et al., 2001*). Due to the importance of protein synthesis in the regenerative growth of axons, we examined the role of eIF2Bε in optic never regeneration. Phosphorylation of eIF2Bε at Ser535 in RGCs was significantly higher in adult mice in comparison with that in the developing retina (*Figure 5—figure supplement 1*), indicating that eIF2Bε-controlled protein translation is maintained at a low basal level in adult RGCs. Optic nerve injury alone did not activate eIF2Bε as the phosphorylation of eIF2Bε was detected at similar levels to uninjured retinas after ONC (*Figure 5A*). By contrast, phospho-eIF2Bε (Ser535) immunoreactivity was markedly reduced in *Gsk3b*-deleted RGCs in the presence or absence of axon injury (*Figure 5A–B*), indicating increased eIF2Bε activity for protein translation. *Gsk3a* deletion did not change eIF2Bε phosphorylation (*Figure 5—figure supplement 2*), suggesting that GSK3α may not regulate eIF2Bε activity in RGCs. To determine the role of eIF2Bε in GSK3β-mediated axon regeneration, we inhibited the activity of eIF2Bε in *Gsk3b*-deleted RGCs by co-injection of AAV-eIF2Bε E572A, a dominant negative mutant of eIF2Bε that lacks catalytic activity but is capable of forming heteropentameric eIF2B complex (*Boesen et al., 2004*; *Wang and Proud, 2008*). Two weeks after ONC, the number of regenerating axons was significantly reduced by AAV-eIF2Bε E572A co-injection (*Figure 5C–D*). To examine whether protein synthesis is involved in eIF2Bε-regulated axon regeneration, we used Alexa Fluor-conjugated OPP incorporation assay to label newly translated proteins. While the basal level of protein translation in RGCs was suppressed by treatment with the protein synthesis inhibitor anisomycin, *Gsk3b* deletion resulted in a significant increase in OPP-labeled new synthesized proteins, which was largely negated by eIF2Bε E572A treatment (*Figure 5E–F*). These results clearly demonstrate that protein synthesis is tightly controlled by GSK3β/eIF2Bε signaling during axon regeneration. However, inhibition of eIF2Bε had no effect on the survival of *Gsk3b*-deleted RGCs (*Figure 5G–H*).

As AKT is further upstream of eIF2Bε, we next examined the effect of eIF2Bε inhibition on AKT-induced axon regeneration. At 2 weeks after ONC, the regenerative response stimulated by AAV-caAKT injection was significantly reduced by co-injection of AAV-eIF2Bε E572A (*Figure 6A–B*), while AKT-induced RGC survival was not perturbed by eIF2Bε E572A inhibition (*Figure 6C–D*). Taken together, our results demonstrate that eIF2Bε is required for both AKT and GSK3β-mediated effects on axon regeneration, but plays a minimal role in RGC survival after axon injury.

## Activation of eIF2Bε promotes axon regeneration

To investigate whether activation of eIF2Bε alone is sufficient to promote axon regeneration, we injected AAV-eIF2Bε S535A, a constitutively active mutant of eIF2Bε that is resistant to GSK3β phosphorylation (*Pap and Cooper, 2002*), in the adult wild-type retina. As expected, expression of

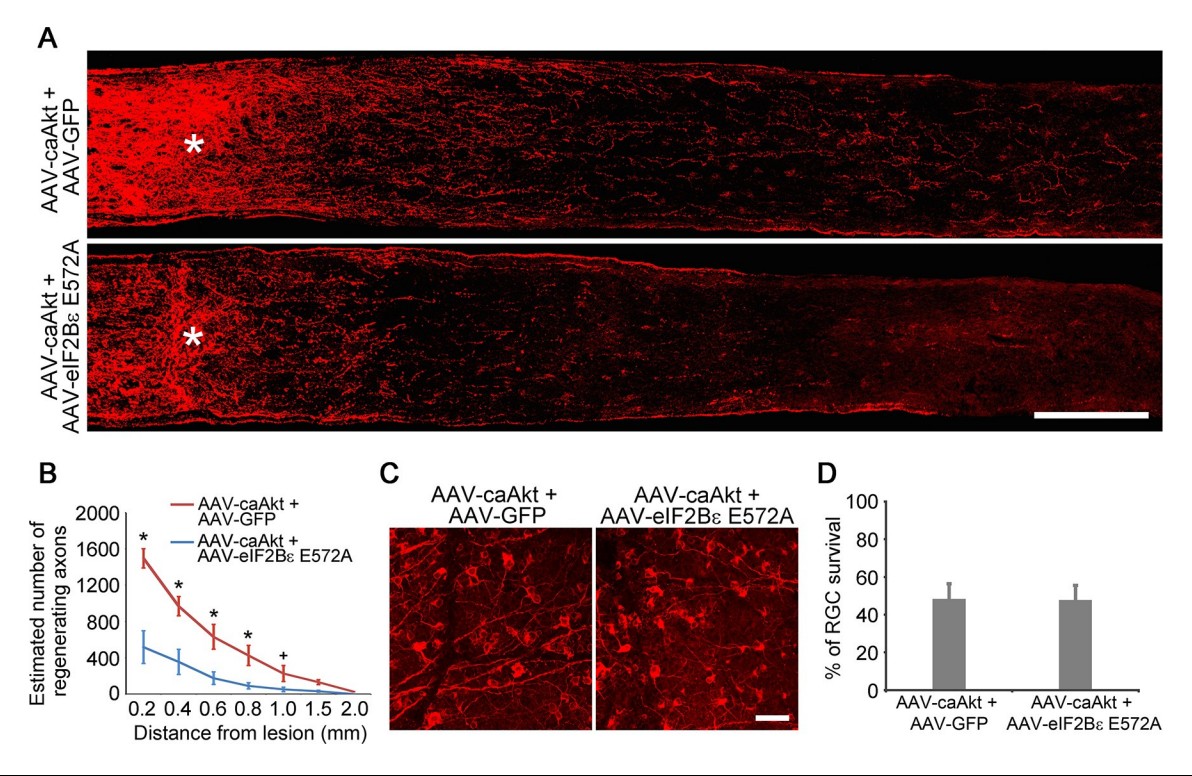

**Figure 6.** eIF2Bε is required for AKT-induced axon regeneration. (**A**) Confocal images of optic nerve sections showing regenerating axons labeled by anti-GAP43 immunohistochemistry at 2 weeks after injury from AAV-caAKT + AAV-GFP or AAV-caAkt + AAV-eIF2Bε E572A injected eyes. *, crush site. Scale bar, 200 μm. (**B**) Quantification of regenerating axons from AAV-caAKT + AAV-GFP or AAV-caAkt + AAV-eIF2Bε E572A injected eyes at different distances distal to the lesion site. Data are presented as mean ± s.d., n=5 per group. *p<0.05, +p<0.05, Two-way ANOVA with Bonferroni *post hoc* test. (**C**) Confocal images of retinal whole-mounts showing surviving Tuj1+ RGCs at 2 weeks after optic nerve injury. Scale bar, 50 μm. (**D**) Quantification of RGC survival at 2 weeks after injury, expressed as a percentage of total Tuj1+ RGCs in the uninjured retina. Data are presented as mean ± s.d., n=5 per group.

eIF2Bε S535A significantly increased protein synthesis in RGCs compared with AAV-GFP injection as a control (*Figure 7A–B*). At 2 weeks after ONC, we observed many regenerating axon fibers extending past the crush site in the eIF2Bε S535A treated retinas (*Figure 7C–D*). The regenerative effect induced by eIF2Bε activation was reduced by protein synthesis inhibition with anisomycin treatment, but was not affected by mTORC1 inhibition with rapamycin treatment (*Figure 7—figure supplement 1*), indicating that mTORC1 signaling plays a minimal role in eIF2Bε-induced axon regeneration. We further assessed whether activation of eIF2Bε affects neuronal survival at 2 weeks after ONC, and found that AAV-eIF2Bε S535A injection did not change the rate of RGC survival in comparison with the control injection (*Figure 7E–F*). These results demonstrate that activation of eIF2Bε promotes axon regeneration with minimal effects on RGC survival.

## Discussion

Unlike the PNS neurons that are capable of regenerating axons after nerve injury, neurons in the adult mammalian CNS fail to regenerate axons after nerve damages. Both the inhibitory extrinsic environment and the diminished intrinsic regenerative capability in the adult CNS contribute to the poor regeneration of axons after injury. Removal of the inhibitory influences in the extracellular environment is insufficient to stimulate a major regenerative response. On the other hand, activation of intrinsic signaling pathways within CNS neurons has yielded encouraging results in promoting axon regeneration (*de Lima et al., 2012*; *Kurimoto et al., 2010*; *Liu et al., 2011*; *Park et al., 2008*; *Smith et al., 2009*; *Sun et al., 2011*; *Yang et al., 2014*). Deletion of *Pten*, a negative regulator of the PI3K/AKT pathway, stimulated robust axon regeneration (*Park et al., 2008*). We found that AKT

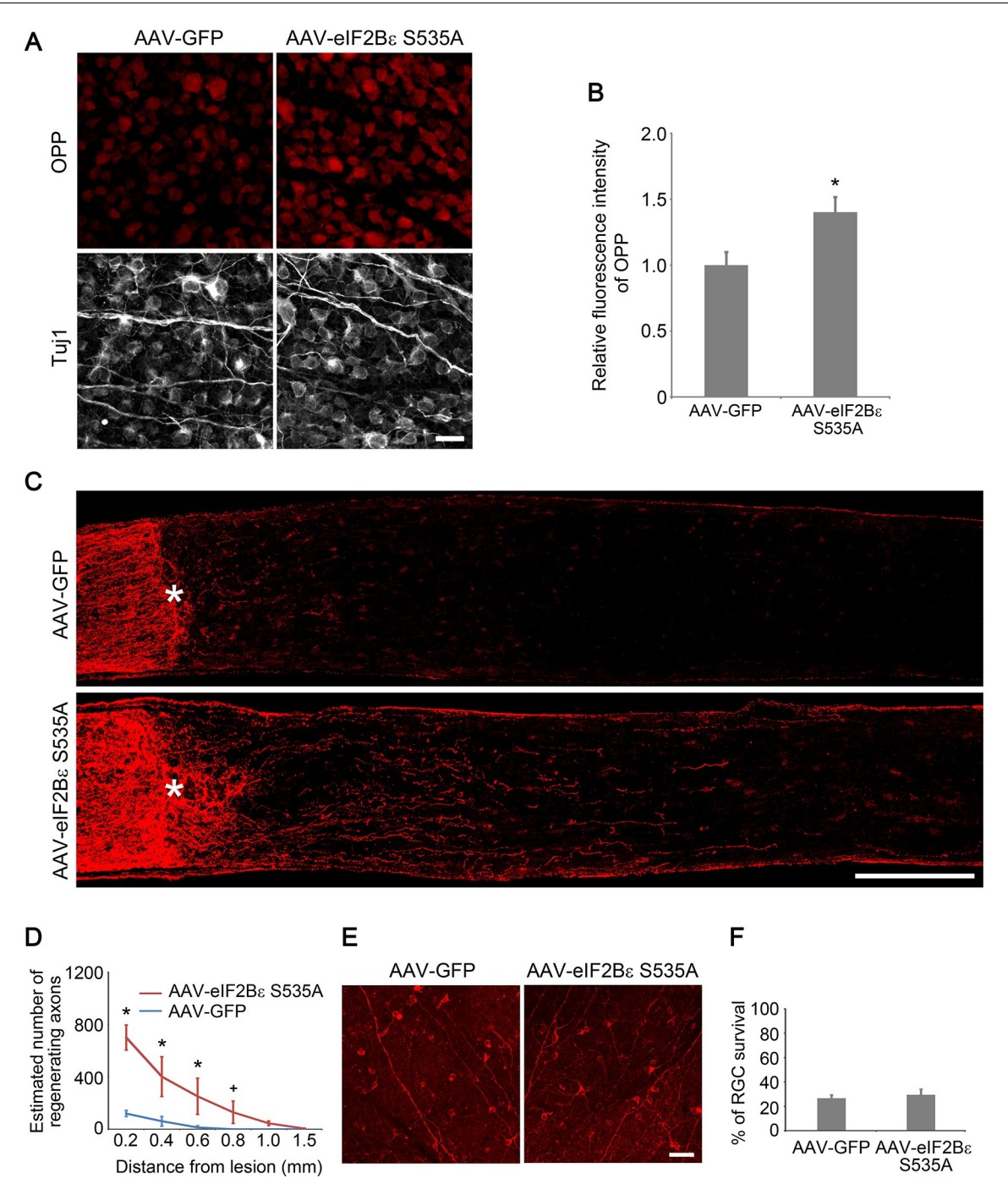

**Figure 7.** eIF2Bε activation promotes axon regeneration. (**A**) OPP Alexa Fluor 594 protein synthesis assay in retinal whole-mounts from AAV-GFP or AAV-eIF2Bε S535A injected eyes. Scale bar, 25 μm. (**B**) Quantification of OPP fluorescence intensity. Data are presented as mean ± s.d., n=5 per group. *p<0.01, Student's *t* test. (**C**) Confocal images of optic nerve sections showing regenerating axons labeled by anti-GAP43 immunohistochemistry at 2 weeks after injury from wild-type mice injected with AAV-GFP or AAV-eIF2Bε S535A. *, crush site. Scale bar, 200 μm. (**D**) Quantification of regenerating axons from retinas injected with AAV-GFP or AAV-eIF2Bε S535A at different distances distal to the lesion site. Data are presented as mean ± s.d., n=5 per group. *p<0.01, +p<0.05, Two-way ANOVA with Bonferroni *post hoc* test. (**E**) Confocal images of retinal whole-mounts showing surviving Tuj1+ RGCs at 2 weeks after optic nerve injury. Scale bar, 50 μm. (**F**) Quantification of RGC survival at 2 weeks after injury, expressed as a percentage of total Tuj1+ RGCs in the uninjured retina. Data are presented as mean ± s.d., n=5 per group.

The following figure supplements are available for figure 7:

*Figure 7 continued*

**Figure supplement 1.** eIF2Bε-induced axon regeneration is sensitive to protein synthesis inhibition but not to mTORC1 inhibition.

**Figure supplement 2.** A schematic illustration of GSK3β/eIF2Bε and mTORC1 signaling in AKT-induced CNS axon regeneration.

was phosphorylated at Ser473 and thus activated in the *Pten*-deleted RGCs. Significantly, expression of a constitutively active myristoylated AKT promoted both axon regeneration and RGC survival. AKT regulates a number of diverging downstream signaling pathways through phosphorylation of diverse cellular targets (*Manning and Cantley, 2007*). Previous studies have established that the mTORC1 pathway is an important downstream signaling in mediating *Pten* deletion-induced axon regeneration, as reactivation of this pathway by TSC1 deletion or S6K1 activation effectively promotes axon regeneration (*Park et al., 2008*; *Yang et al., 2014*). However, mechanisms other than the mTORC1 pathway downstream of *Pten* deletion have not been elucidated.

Several studies have demonstrated the role of GSK3 signaling in developmental axon growth (*Hur et al., 2011*; *Hur and Zhou, 2010*; *Kim et al., 2006*). In the adult PNS, GSK3 signaling regulates mammalian sensory axon regeneration by inducing the expression of Smad1 (*Saijilafu et al., 2013*). In the spinal cord injury model, neuronal deletion of *Gsk3b* enhances dorsal column axon regeneration via CRMP2-regulated microtubule dynamics, specifically in the growth cone (*Liz et al., 2014*). Here, using the optic nerve injury model, we have demonstrated that GSK3β plays a pivotal role in regulating CNS axon regeneration. *Pten* deletion or AKT activation led to phosphorylation of GSK3β at the site of Ser9, resulting in inactivation of its kinase activity in RGCs. Expression of GSK3β S9A, a kinase active mutant of GSK3β, significantly reduced axon regeneration induced by AKT activation. Deletion of *Gsk3b* or inactivation of GSK3β with the expression of GSK3β K85A, a kinase dead mutant of GSK3β, was sufficient to promote axon regeneration. However, GSK3α did not have these same effects. These results are in line with studies showing GSK3 isoform-specific effects in developmental axon growth (*Chen et al., 2012*). Indeed GSK3α and GSK3β have been shown to act on different substrates in the brain (*Soutar et al., 2010*). We did not assess the results of complete elimination of GSK3 activity, which has been associated with the suppression of axon growth in developmental studies (*Kim et al., 2006*). Interestingly, unlike AKT activation that promoted both axon regeneration and RGC survival, GSK3 inactivation did not promote RGC survival, suggesting that mTORC1 and/or other unidentified factors downstream of AKT are involved in promoting the survival of RGCs.

How might GSK3β signaling regulate axon regeneration? eIF2Bε, the largest catalytic subunit of eIF2B, plays an important role in initiating protein translation in all eukaryotic cells (*Pavitt, 2005*). The activity of eIF2Bε is inhibited by GSK3β phosphorylation. In response to insulin, GSK3β kinase activity is inhibited, eIF2Bε is dephosphorylated and thus more active, leading to an enhancement of protein translation. In the CNS, eIF2Bε-controlled protein translation is required for the development and maintenance of brain white matter, which is composed of bundles of myelinated axons (*Fogli and Boespflug-Tanguy, 2006*; *Geva et al., 2010*). We found that eIF2Bε is a key downstream target of GSK3β to regulate axon regeneration after optic nerve injury. Inhibition of eIF2Bε with the dominant negative mutant eIF2Bε E572A significantly reduced axon regeneration induced by *Gsk3b* deletion or AKT activation. More importantly, activation of eIF2Bε with the constitutively active mutant eIF2Bε S535A was sufficiently effective to promote axon regeneration, emphasizing the importance of protein synthesis as a major neuronal intrinsic mechanism in CNS axon regeneration.

Protein synthesis is primarily regulated at the initiation phase, which involves binding of the methionyl–transfer RNA (Met–tRNA$_i^{Met}$) to the small 40s ribosomal subunit to form the 43 s pre-initiation complex, which then binds to an mRNA to form the 48 s pre-initiation complex that scans the AUG start codon. Each of these steps in translation initiation is facilitated by proteins referred to as eukaryotic initiation factors eIFs (*Klann and Dever, 2004*). The tRNA$_i^{Met}$ binding to the 40 s ribosomal subunit requires the exchange of GTP-bound for GDP-bound eIF2, which is catalyzed by the guanine-nucleotide exchange factor eIF2B. Therefore, GSK3β-mediated eIF2Bε activation enhances the recycling of eIF2 for further rounds of translational initiation, constituting a critical regulatory mechanism to control the intrinsic regenerative ability of RGCs after nerve injury. The 5′ cap-binding protein eIF4E is required for the binding of the 43 s pre-initiation complex to most eukaryotic

mRNAs. mTORC1 activation enhances protein synthesis through phosphorylation of 4E-BP (translation initiation factor 4E-binding protein), a negative regulator of eIF4E. However, 4E-BP inhibition is not sufficient to promote axon regeneration as co-deletion of 4E-BP1/2 in RGCs does not stimulate optic nerve regeneration. Interestingly, activation of S6K1, another downstream factor of mTORC1, promotes RGC axon regeneration (*Yang et al., 2014*). S6K1 activation promotes the translation of 5' TOP (terminal oligopyrimidine tract) mRNAs, which encode exclusively components of the translation machinery including ribosomal proteins, elongation factors, and poly(A)-binding protein (PABP) (*Hay and Sonenberg, 2004*). It appears that mTORC1 and GSK3β signaling, although acting independently to regulate protein translation downstream of AKT, converge on a common translation regulatory mechanism for axon regeneration.

In summary, our results provide evidence that the AKT-GSK3β-eIF2Bε signaling module plays a central role in determining the intrinsic axonal growth ability of mature CNS neurons (*Figure 7—figure supplement 2*). Identifying various signaling pathways may enable combinatorial treatment to promote axon regeneration after CNS injury.

# Materials and methods

## Animals

C57BL/6 mice were purchased from The Jackson Laboratory (Bar Harbor, Maine). $Pten^{f/f}$ mice were obtained from Dr. William Cafferty's laboratory (Yale University, New Haven, USA). $Gsk3a^{-/-}$ and $Gsk3b^{f/f}$ mice were kindly provided by Dr. James R Woodgett (McMaster University, Hamilton, Ontario, Canada). All studies adhered to the procedures consistent with animal protocols approved by the IACUC at Yale University.

## AAV plasmid vectors and AAV preparation

pAAV-Cre and pAAV-GFP plasmids were kindly provided by Dr. Kevin Park (University of Miami). For plasmids construction, protein-coding region in pAAV-GFP was replaced by the coding sequence of caAKT (myrAkt delta4-129, Addgene plasmid # 10841, Cambridge, MA), GSK3β S9A (obtained from Dr. Marc B Hershenson, University of Michigan), GSK3β K85A (Addgene plasmid # 14755), eIF2Bε S535A (obtained from Dr. Geoffrey M Cooper, Boston University) and eIF2Bε E572A. pAAV2-RC (Stratagene, La Jolla, CA) and the Helper plasmid were used for co-transfection in HEK293T cells. Discontinuous iodixanol gradient ultracentrifugation was used to purify AAV. AAV titers, determined by real-time PCR, were in the range of 1–5 x $10^{12}$ genome copies per milliliter.

## Intravitreal injection and optic nerve crush

Mice were anesthetized with a mix of ketamine (100 mg/kg) and xylazine (10 mg/kg) by intraperitoneal injection. For intravitreal injection, the micropipette was inserted just behind the ora serrata, and 1 µl of AAV solution was injected in the vitreous body. Two weeks after viral injection, optic nerve was exposed and crushed intraorbitally with jeweler's forceps for 5 s approximately 1 mm behind the optic disc. Eye ointment was applied to protect the cornea after surgery.

## Immunohistochemistry and fluorescence microscopy

Eyes with the attached optic nerve segment, surgically removed from perfused mice, were post-fixed in 4% PFA. Retinas were dissected out for either whole-mount preparations or cryosections. The optic nerve was separated from the eye and cut longitudinally with a Leica cryostat. Retinal wholemounts or sections were blocked in the staining buffer containing 5% normal donkey serum and 0.1% Triton X-100 in PBS for 1 hr before incubation with primary antibodies. Primary antibodies used: Tuj1 (Covance, 1:500, Princeton, NJ), PTEN (Cell Signaling Technology, 1:250, Danvers, MA), p-AKT (Cell Signaling Technology, 1:200), AKT (Cell Signaling Technology, 1:250), p-GSK3α (Abcam, 1:250, UK), p-GSK3β (Cell Signaling Technology, 1:400), GSK3b (Cell Signaling Technology, 1:250), p-S6 (Cell Signaling Technology, 1:200), p-eIF2Bε (EMD Millipore, 1:300, Billerica, MA), GAP43 (obtained from Dr. Larry Benowitz, 1:500), CSPG (Sigma, 1:200, St. Louis, MO), and GFAP (Cell Signaling Technology, 1:200). Secondary antibodies used: DyLight Cy3/594/647-conjugated AffiniPure antibodies (Jackson ImmunoResearch, 1:500, West Grove, PA). Confocal images were acquired using

a Zeiss LSM 510 EXCITER microscope. Fluorescence channel colors were switched for co-localization studies if necessary. Images were analyzed and organized using ImageJ and Photoshop.

## Quantification of RGC survival and axon regeneration

For RGC counting, retinal whole-mounts were immunostained with Tuj1 antibody, and 8–12 fields (321 x 321 µm) were randomly sampled from the peripheral regions of each retina. Regenerating axons was quantified by counting the number of GAP43 positive axons at different distance from the crush site in four sections per nerve, as described previously (*Park et al., 2008*). The cross-sectional width of the nerve was measured at the point at which the counts were taken and was used to calculate the number of axons per millimeter of the width of the nerve. The number of axons per millimeter was then averaged over all sections. $\Sigma a_d$, the total number of axons extending distance $d$ in a nerve having a radius of $r$, was estimated by summing over all sections having a thickness $t$: $Sa_d = \pi r^2$ x [average axons/mm]/$t$.

## Rapamycin and anisomycin administration

Rapamycin was administered as previously described (*Park et al., 2008*). Rapamycin (LC Laboratories, Woburn, MA) was dissolved at 20 mg/ml in ethanol for stock. Before each administration, rapamycin was diluted in 5% Tween 80, 5% polyethylene glycol 400 (1.0 mg/ml) in PBS. Rapamycin at 6 mg/kg or the vehicle was injected intraperitoneally once every 2 days after AAV injection. Anisomycin (Cayman Chemical, Ann Arbor, MI) was dissolved in DMSO to prepare a stock solution and diluted in PBS before each administration. Anisomycin (30 mg/kg body weight) or vehicle was injected subcutaneously daily after optic nerve crush. To inhibit OPP incorporation, anisomycin (50 µg/ml) was injected intravitreally 1 hr before and together with OPP administration.

## Protein synthesis analysis and imaging

For new protein synthesis analysis and imaging, we used Click-iT Plus OPP Alexa Fluor 594 Protein Synthesis Assay Kit (Life Technologies, Carlsbad, CA). In this assay, OPP (O-propargyl-puromycin) is efficiently incorporated into newly translated proteins that can be detected by fluorescently labeled Alexa Fluor dye. One microliter of OPP reagent (200 µM in PBS) was injected into the vitreous body per eye. One hour after injection, eyes were dissected out after perfusion and were post-fixed in 4% PFA for 4 hr. OPP detection was performed according to the manufacturer's protocol. Retinas were subsequently immunostained with the Tuj1 antibody to label RGCs.

## Acknowledgements

We thank Dr. James R Woodgett (McMaster University, Hamilton, Ontario, Canada) for providing *Gsk3b^{f/f}* and *Gsk3a* knockout mice. We thank Dr. Kevin Park (University of Miami) for providing pAAV-Cre and pAAV-GFP plasmids; Dr. Marc B Hershenson (University of Michigan) for providing GSK3β S9A plasmid; Dr. Geoffrey M Cooper (Boston University) for providing eIF2Bε S535A and eIF2Bε plasmids; Dr. Larry Benowitz (Children's Hospital, Harvard Medical School) for providing GAP43 antibody. This work was supported by an unrestricted grant from Research to Prevent Blindness to Yale University.

## Additional information

### Funding

| Funder | Grant reference number | Author |
|---|---|---|
| Yale University | New Faculty Award | Bo Chen |
| Research to Prevent Blindness | Unrestricted fund to the Yale Eye Center | Bo Chen |

The funders had no role in study design, data collection and interpretation, or the decision to submit the work for publication.

## Author contributions

XG, Conception and design, Acquisition of data, Analysis and interpretation of data, Drafting or revising the article; WDS, Discussion of the project design and participation in drafting the manuscript; BC, Conception and design, Analysis and interpretation of data, Drafting or revising the article

## Author ORCIDs

Bo Chen, http://orcid.org/0000-0003-2768-9007

## Ethics

Animal experimentation: This study was performed in strict accordance with the recommendations in the Guide for the Care and Use of Laboratory Animals of the National Institutes of Health. All of the animals were handled according to approved institutional animal care and use committee (IACUC) protocols (#2013-11389) of Yale University. Yale University has an approved Animal Welfare Assurance (#A3230-01) on file with the NIH office of Laboratory Animal Welfare. All surgery was performed under ketamine anesthesia, and every effort was made to minimize suffering.

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
