## [Decision Letter]

Thank you for submitting your work entitled "GSK3β regulates AKT-induced CNS axon regeneration through an eIF2Bε-dependent, mTORC1-independent signaling pathway" for consideration by *eLife*. Your article has been reviewed by three peer reviewers: Gary Bassell, Fengquan Zhou and a member of our Board of Reviewing Editors. The evaluation has been overseen by the Reviewing Editor and K VijayRaghavan as the Senior Editor.

The reviewers have discussed the reviews with one another and the Reviewing Editor has drafted this decision to help you prepare a revised submission.

Summary:

Guo et al. show that the GSK3β-eIF2Bε pathway, downstream of PTEN-AKT, is important for CNS axon regeneration in a model of optic nerve crush (ONC). They demonstrate that upon PTEN genetic ablation (which has been demonstrated to result in CNS axonal outgrowth), there is activation of AKT, downstream phosphorylation and inactivation of GSK3β, and consequently, activation (disinhibition) of eIF2Bε. The activation of this pathway results in significant axon regeneration, accompanied by increased protein synthesis. The identification of the GSK3β-eIF2Bε pathway as relevant for axonal regeneration expands the scope of potential targets to improve axonal regeneration following CNS nerve injury. The study has effectively integrated use of viral vectors to express constitutively active and dominant negative forms of several pathway components, together with Cre-mediated knockout, and systematically show that regeneration is induced in a GSK3β independent of mTORC1 signaling. The data are convincing and the manuscript is presented in a logical and straightforward manner. The experiments were carefully designed and executed, and the results are technically sound.

Essential revisions:

1) The statistical methods used through the bulk of the paper are inappropriate. Figure. 1B, 1D; Figure 2; Figure 3; Figure 5; Figure 6; Figure 1—figure supplement 1; Figure 2—figure supplement 1; Figure 3—figure supplement 1; and Figure 4—figure supplement 1 need to be analyzed by a 2-way ANOVA with appropriate post-hoc tests. It is inappropriate to use repeated t-tests. In Figure 4—figure supplement 1 there are also no asterisks on the graph, although they are indicated in the legend.

2) The major method of analysis for pathway activation is the scoring of percent positive cells, which can be subjective and highly variable. It would be helpful to show fluorescence intensity measurements for a couple of the manipulations.

3) The OPP labeling method for protein synthesis is in need of a negative control experiments using anisomycin to block the incorporation (Figure 5).

4) One of the central elements of this report is the emphasis on the independence of the GSK3β-eIF2Bε pathway, downstream of AKT, from activation of mTORC1 in the stimulation of axon outgrowth following injury. In fact, it appears that unlike mTORC1 activation, the GSK3β-eIF2Bε pathway does not affect RGC survival following ONC (Figure 2, Figure 3, Figure 5, Figure 6 and Figure 7) while the data are quite convincing to suggest that both branches affect axonal regeneration. When rapamycin is administered (in the presence of a constitutively active AKT), axonal regeneration is almost completely inhibited (Figure 1—figure supplement 1, panels A and B), but the authors claim that this inhibition is partial. Without a direct comparison between the AAV-GFP, AAV-caAKT, and AAV-caAKT+Rapamycin groups (Figure 1) this claim is weak. The same rationale applies to the statement on the effects on RGC survival (presented in Figure 1—figure supplement 1, panel D).

5) The magnitude of the regenerative effect that is dependent on GSK3β phosphorylation by AKT (Figure 2) is comparable to that of rapamycin (Figure 1—figure supplement 1, panels A and B). Nonetheless, when GSK3β is inactivated (Figure 3 and Figure 5), the regenerative effect appears smaller than that resulting from AKT constitutive activation (Figure 1, Figure 2 and Figure 6) (although a direct comparison of these effects is missing). It therefore appears important to acknowledge that both branches, although independent, result in the stimulation of mRNA translation initiation, thus converging on a common mechanism for axonal regeneration. One experiment that is lacking is the demonstration that protein synthesis inhibitors block the regenerative effect of the GSK3β-eIF2Bε signaling axis.

6) Another aspect in this study that was not addressed is whether the GSK3β-eIF2Bε pathway is indeed modulated by ONC or perhaps other models of CNS nerve injury. In this regard, mTORC1 activity, which is developmentally decreased, is reduced by ONC in RGCs (Park et al. 2008); it would be important to show whether this is the case for the AKT-GSK3β-eIF2Bε pathway.

7) Relevant to comment #4, is the interesting finding that knocking out GSK3α had no promoting effect. Although the authors mentioned briefly in the Discussion that GSK3α might regulate different substrates from GSK3β, there was no experimental evidence provided. For instance, did AKT activation also inactivate GSK3α in RGCs (Will caAKT lead to increased level of GSK3α-S21 phosphorylation)? How did GSK3α knockout affect the phosphorylation of eIF2Bε (Is eIF2Bε phosphorylation reduced in GSK3α knockout mice)? Answers to these questions would greatly improve the study. These experiments could be performed in a short time.

8) The OPP labeling in Figure 5 and Figure 7 need to be quantified.

9) Some discussion about the difference between eIF2Bε and eIF4E mediated protein synthesis would be helpful.

10) In terms of the role of GSK3 signaling in PNS axon regeneration, one study (a) showed that GSK3α/β Ser21/9-Ala double knock-in mice had no obvious phenotype in sensory axon regeneration. The finding was different from a later study (b). Both studies need to be cited. In addition, an earlier study (c) has shown that GSK3β knockout could enhance dorsal column axon regeneration in the spinal cord via CRMP2, which should also be cited.

a) Zhang BY, et al. (2014) AKT-independent GSK3 inactivation downstream of PI3K signaling regulates mammalian axon regeneration. Biochem Biophys Res Commun 443:743-748.b) Gobrecht P, Leibinger M, Andreadaki A, Fischer D (2014) Sustained GSK3 activity markedly facilitates nerve regeneration. Nat Commun 5:4561.c) Liz MA, et al. (2014) Neuronal deletion of GSK3β increases microtubule speed in the growth cone and enhances axon regeneration via CRMP-2 and independently of MAP1B and CLASP2. BMC Biol 12:47.

---

## [Author Response]

*1) The statistical methods used through the bulk of the paper are inappropriate. Figure 1, Figure 2, Figure 3, Figure 5, Figure 6, Figure 1—figure supplement 1, Figure 2—figure supplement 1, Figure 3—figure supplement 1, Figure 4—figure supplement 1 need to be analyzed by a 2-way ANOVA with appropriate post-hoc tests. It is inappropriate to use repeated t-tests. In Figure 4—figure supplement 1 there are also no asterisks on the graph, although they are indicated in the legend.*

We thank the reviewers for their suggestions to use statistical methods that are more appropriate for the data analysis. Now we have used two-way ANOVA with Bonferroni post hoc test for Figure 1; Figure 2; Figure 3; Figure 5; Figure 6; Figure 1—figure supplement 1; Figure 2—figure supplement 1; Figure 3—figure supplement 1; and Figure 4—figure supplement 1 in the initial submission, and they correspond to Figure 1; Figure 2; Figure 3; Figure 5; Figure 6; Figure 1—figure supplement 2; Figure 2—figure supplement 2; Figure 3—figure supplement 2; and Figure 4—figure supplement 1 respectively in the revised manuscript. In addition, asterisks indicating statistical difference have been added to Figure 4—figure supplement 1 in the revised manuscript.

*2) The major method of analysis for pathway activation is the scoring of percent positive cells, which can be subjective and highly variable. It would be helpful to show fluorescence intensity measurements for a couple of the manipulations.*

We agree with the reviewers that scoring the percentage of positively immunostained cells can be subjective and highly variable from one experiment to another. In the revised manuscript, we have used measurements of immunofluorescence intensity for the analysis of pathway activation for phospho-AKT, phospho-GSK3β, phospho-S6, and phospho-eIF2Bε.

*3) The OPP labeling method for protein synthesis is in need of a negative control experiments using anisomycin to block the incorporation (Figure 5).*

As advised by the reviewers, we used anisomycin, injected intravitreally 1 hour before and together with OPP administration, to block protein synthesis, and observed a significant decrease in OPP incorporation. The added negative control experiment is shown in Figure 5 in the revised manuscript.

*4) One of the central elements of this report is the emphasis on the independence of the GSK3β-eIF2Bε pathway, downstream of AKT, from activation of mTORC1 in the stimulation of axon outgrowth following injury. In fact, it appears that unlike mTORC1 activation, the GSK3β-eIF2Bε pathway does not affect RGC survival following ONC (Figure 2, Figure 3, Figure 5, Figure 6 and Figure 7) while the data are quite convincing to suggest that both branches affect axonal regeneration. When rapamycin is administered (in the presence of a constitutively active AKT), axonal regeneration is almost completely inhibited (Figure 1—figure supplement 1, panels A and B), but the authors claim that this inhibition is partial. Without a direct comparison between the AAV-GFP, AAV-caAKT, and AAV-caAKT+Rapamycin groups (Figure 1) this claim is weak. The same rationale applies to the statement on the effects on RGC survival (presented in Figure 1—figure supplement 1, panel D).*

We agree with the reviewers that a direct comparison between the AAV-GFP, AAV-caAKT, and AAV-caAKT+Rapamycin treatment groups is needed to make a convincing conclusion regarding the role of mTORC1 in the AKT-mediated axon regeneration and RGC survival. In the revised manuscript, a direct comparison of these 3 treatment groups has been made to assess axon regeneration (Figure 1—figure supplement 2) and RGC survival (Figure 1—figure supplement 2). We found that inhibition of mTORC1 by rapamycin treatment significantly, but not completely, reduced AKT activation-induced axon regeneration. On the other hand, AKT activation-induced RGC survival was almost completely suppressed by rapamycin treatment.

*5) The magnitude of the regenerative effect that is dependent on GSK3β phosphorylation by AKT (Figure 2) is comparable to that of rapamycin (Figure 1—figure supplement 1, panels A and B). Nonetheless, when GSK3β is inactivated (Figure 3 and Figure 5), the regenerative effect appears smaller than that resulting from AKT constitutive activation (Figure 1, Figure 2 and Figure 6) (although a direct comparison of these effects is missing). It therefore appears important to acknowledge that both branches, although independent, result in the stimulation of mRNA translation initiation, thus converging on a common mechanism for axonal regeneration. One experiment that is lacking is the demonstration that protein synthesis inhibitors block the regenerative effect of the GSK3β-eIF2Bε signaling axis.*

The reviewers were correct that GSK3β inactivation, either by Cre-mediated deletion or expression of a kinase dead mutant GSK3β K85A, generated a smaller regenerative effect in comparison to AKT constitutive activation, and this is because both mTORC1 and GSK3β-eIF2Bε signaling branches act independently downstream of AKT to stimulate mRNA translation initiation and promote axon regeneration. To demonstrate that protein synthesis is a common regulatory mechanism for axon regeneration, we have performed the experiments suggested by the reviewers to inhibit protein synthesis using anisomycin and assess GSK3β-deletion induced axon regeneration (Figure 4—figure supplement 1) and elF2Bε activation induced axon regeneration (Figure 7—figure supplement 1), and found that anisomycin treatment significantly reduced the regenerative effect of the GSK3β-eIF2Bε signaling axis. In addition, we have also assessed the effect of GSK3β inactivation (Figure 4—figure supplement 1) or eIF2Bε activation (Figure 7—figure supplement 1) on RGC survival when protein synthesis was inhibited by anisomycin treatment, and found that the GSK3β-eIF2Bε signaling axis did not affect RGC survival.

*6) Another aspect in this study that was not addressed is whether the GSK3β-eIF2Bε pathway is indeed modulated by ONC or perhaps other models of CNS nerve injury. In this regard, mTORC1 activity, which is developmentally decreased, is reduced by ONC in RGCs (Park et al. 2008); it would be important to show whether this is the case for the AKT-GSK3β-eIF2Bε pathway.*

To assess whether the AKT-GSK3β-eIF2Bε pathway is developmentally regulated, we analyzed the activity of AKT (Figure 1—figure supplement 1), GSK3β (Figure 2—figure supplement 1), and eIF2Bε (Figure 5—figure supplement 1) at postnatal day 7, 21, and 60. Indeed, we found that the AKT-GSK3β-eIF2Bε signaling pathway is developmentally regulated such that this pathway was activated in the developing retina at postnatal day 7 and downregulated in the adult age at postnatal day 21 and 60.

To assess whether the AKT-GSK3β-eIF2Bε pathway is modulated by optic nerve injury, we collect retinas without injury 1or 3 days after ONC. However, we did not observe a significant change in the phosphorylation of AKT (Figure 1), GSK3β (Figure 2), or eIF2Bε (Figure 5) in comparison to uninjured animals.

*7) Relevant to comment #4, is the interesting finding that knocking out GSK3α had no promoting effect. Although the authors mentioned briefly in the Discussion that GSK3α might regulate different substrates from GSK3β, there was no experimental evidence provided. For instance, did AKT activation also inactivate GSK3α in RGCs (Will caAKT lead to increased level of GSK3α-S21 phosphorylation)? How did GSK3α knockout affect the phosphorylation of eIF2Bε (Is eIF2Bε phosphorylation reduced in GSK3α knockout mice)? Answers to these questions would greatly improve the study. These experiments could be performed in a short time.*

The reviewers raised important questions regarding the role of GSK3α in RGC axon regeneration since knocking out GSK3α had no promoting effect. To address these questions, we analyzed the activity of GSK3α by performing phospho-GSK3α-Ser21 immunohistochemistry following AAV-caAKT injection, using AAV-GFP injection as a control, and found that AKT activation inactivated GSK3α, either with or without optic nerve injury (Figure 3—figure supplement 3). We have also analyzed the phosphorylation of eIF2Bε in the GSK3α knockout mice (Figure 5—figure supplement 2). However, the phospho-eIF2Bε level was not affected by GSK3α deletion. Our results indicate that both GSK3α and GSK3β are phosphorylated and thus inactivated by AKT, but only GSK3β regulates the activity of eIF2Bε. This may explain why no axon regeneration was observed after optic nerve injury in GSK3α knockout mice.

*8) The OPP labeling in Figure 5 and Figure 7 need to be quantified.*

We have quantified the OPP labeling in Figure 5 and Figure 7 by measuring the OPP fluorescence intensity. And new figure panels, Figure 5 and Figure 7 respectively, have been added to show quantified results of the OPP labeling in the revised manuscript.

*9) Some discussion about the difference between eIF2Bε and eIF4E mediated protein synthesis would be helpful.*

While both eIF2Bε and eIF4E play an important role in protein translation initiation, they mediate different steps in the assembly of the translation initiation complex. A detailed discussion has been added in the revised manuscript (Discussion, paragraphs three to five), to describe the differential roles of eIF2Bε and eIF4E in mRNA translation, and how their activity is regulated by GSK3β and mTORC1 in the context of axon regeneration.

*10) In terms of the role of GSK3 signaling in PNS axon regeneration, one study (a) showed that GSK3α/*β *Ser21/9-Ala double knock-in mice had no obvious phenotype in sensory axon regeneration. The finding was different from a later study (b). Both studies need to be cited. In addition, an earlier study (c) has shown that GSK3β knockout could enhance dorsal column axon regeneration in the spinal cord via CRMP2, which should also be cited.*

*a) Zhang BY, et al. (2014) AKT-independent GSK3 inactivation downstream of PI3K signaling regulates mammalian axon regeneration. Biochem Biophys Res Commun 443:743-748.b) Gobrecht P, Leibinger M, Andreadaki A, Fischer D (2014) Sustained GSK3 activity markedly facilitates nerve regeneration. Nat Commun 5:4561.c) Liz MA, et al. (2014) Neuronal deletion of GSK3β increases microtubule speed in the growth cone and enhances axon regeneration via CRMP-2 and independently of MAP1B and CLASP2. BMC Biol 12:47.*

We have noticed the different results regarding the role of GSK3 in PNS axon regeneration using the GSK3α-S21A/GSK3β-S9A double knock-in mice. One group (Gobrecht P, et al. 2014) showed that constitutive activation of GSK3 resulted in markedly increased axon regeneration, while the other group (Zhang BY, et al. 2014) observed no obvious effects in sensory axon regeneration. We have cited the two studies on page 5 in the revised manuscript.

Using the spinal cord injury model, Liz MA, et al. showed that neuronal deletion of GSK3β enhances dorsal column axon regeneration via CRMP2-regulated microtubule dynamics, specifically in the growth cone. We have cited this paper in the Discussion section.